# Unconscious Drivers of Consumer Behavior: An Examination of the Effect of Nature–Nurture Interactions on Product Desire

**DOI:** 10.3390/bs14090789

**Published:** 2024-09-07

**Authors:** Jim B. Swaffield, Jesus Sierra Jimenez

**Affiliations:** 1Faculty of Business, Athabasca University, Athabasca, AB T9S 3A3, Canada; 2Faculty of Management, Vancouver Island University, Nanaimo, BC V9R 5S5, Canada; jesus.sierrajimenez@viu.ca

**Keywords:** product signalling, environmental harshness, nature–nurture interaction, adaptive behavior

## Abstract

Both biological and environmental factors can affect consumer behavior. Consumer behavior can also be a product of an interaction between one’s evolved biology and environmental factors. If marketers aim to increase healthy consumption behavior and decrease unhealthy behavior, they need to identify whether the behavior is a product of one’s evolved biology or environmental factors acting in isolation, or if the behavior is a product of a biology–environment interaction. Therefore, the purpose of this study is to examine the effect of biology–environment interactions on product desire. This study comprises two experiments that used a repeated-measures design. The first experiment included 315 females and examined the effect of perceived physical safety, economic well-being, and social support on the desire for beautifying and wealth-signalling products. The second experiment included 314 men and examined the effect of perceived physical safety, economic well-being, and social support on the desire for products that are used to signal wealth and toughness. The results showed that under harsh economic conditions, product desire generally decreased. However, there were significant differences in the amount of decrease between product categories in different environmental conditions.

## 1. Introduction

It is sometimes said, that “People buy products they don’t really need, with money they don’t really have, to impress people they don’t really like” [1]. These behaviors are often seen as irrational, especially when they lead to problems such as compulsive buying behavior, overeating, and the accumulation of excessive debt. Unfortunately, many interventions aimed at reducing problematic consumer behaviors have poor long-term effectiveness. For example, training and counseling services that promote weight loss and healthy eating behavior seldom have a positive long-term effect [2]. Debt counseling services and interventions aimed at abating compulsive buying behavior also have a poor record of long-term effectiveness [3,4].

A possible reason for the ineffectiveness of various interventions may be due to incorrect assumptions regarding the cause of problematic consumer behaviors. Specifically, the effectiveness of interventions aimed at reducing problematic behavior begins with an accurate diagnosis and identification of the cause of the problem. Stated another way, poor problem diagnosis leads to the development of poor interventions.

The purpose of this article and research study is not to address specific problematic consumer behaviors, nor is the purpose to offer intervention advice. But rather, the purpose is to provide a new way of thinking about what drives product desire and problematic consumer behavior. Specifically, this paper posits that consumer behavior is neither a product of nature nor nurture acting in isolation, but rather, consumer behavior is a product of an interaction between one’s nature (biology) and nurture (environmental factors). This nature–nurture interaction is sometimes described through the metaphor, “genetics loads the gun, and the environment pulls the trigger” [5]. For example, humans have an instinctual need for safety. However, the desire for products such as a home security system is dependent upon the perceived level of environmental threats and perceived risk of personal injury.

While prior studies have examined the effect of nature [6,7,8,9] and nurture [8,10,11,12,13,14,15,16,17] on product desire, there is a gap in the research literature that has examined how *nature–nurture interactions* affect product desire. In addition, there is also an absence of research that has examined how *nature–nurture interactions* affect desire for different product categories, such as conspicuous consumption products.

To fill this existing knowledge gap and to advance our understanding of this emerging field of research, we examine the effect of nature–nurture interactions on product desire. This study deepens our understanding of how nature–nurture interactions can mediate and moderate the desire for products that are often used in a conspicuous consumption manner.

Thus, this paper proceeds as follows. In Section 2, we review the prior research that examines the effect of nature or nurture on product desire. Next, in Section 3, we outline the focus of the study and define our hypotheses. In Section 4, Methods, we describe the study participants, how they were recruited, and the procedures for implementing the study. This section also includes an analysis of the results. Finally, this paper concludes with a discussion of the practical and theoretical implications of the study findings.

## 2. Review of Related Studies

### 2.1. The Effect of Nature on Consumer Behavior

Human nature includes one’s evolved biology, genetics, and instincts. These traits have been selected through evolutionary pressures and have helped our early ancestors survive and reproduce [6]. For example, the following seven ‘socially oriented’ traits have been hardwired into humans through an evolutionary process. These evolved traits include the drive to elevate one’s position within their respective social status hierarchy, to evade physical harm, avoid disease, make friends, acquire a mate, retain a mate, and care for and protect one’s family and kin [18].

It is inferred that these traits are a product of our evolved biology because they exist in all modern-day cultures and have existed throughout history. Examples of consumer behaviors that are associated with these socially evolved traits include purchasing, consuming and sharing food, gift giving, buying beautifying products, and the conspicuous display of products that signal wealth and toughness [7].

### 2.2. The Effect of Nurture on Consumer Behavior

In contrast to one’s nature, which is coded into one’s body and brain prior to conception, ‘nurture’ refers to the effect of environmental conditions and stimuli experienced after the point of conception [19]. These environmental forces may be experienced unconsciously or consciously. In addition to the influence of unconscious environmental stimuli, consumer behavior can be influenced through informal conscious observation and direct instruction. Both informal observation and direct instruction can occur in social settings such as the family environment, religious and educational settings, and by participating in social gatherings.

Informal social learning can also occur through the media and by observing commercials and advertisements. Based on the fact that consumers can learn through commercials and advertisements, it is often assumed that problematic behaviors such as excessive food consumption are a product of prior learning [20]. It is also frequently assumed that the solution to preventing problematic consumption behaviors is to block consumers from seeing or hearing advertisements and commercials. For example, in an effort to reduce childhood obesity, governments from around the world have banned, or are in the process of passing legislation to ban the advertising of fast food and junk food to children [21,22,23].

While consumers can learn through marketing and advertising initiatives, we should not infer that advertising in isolation drives consumer behavior or is the cause of problematic consumption behaviors. Likewise, it should not be assumed that consumer education or the removal of marketing and advertising stimuli will have an effect on the reduction in problematic consumption behaviors [2,3,4,24].

### 2.3. The Nature–Nurture Interaction and Consumer Behavior

Researcher Frans de Waal [25] provides an apt analogy that underscores the importance of studying human behavior from an interaction perspective. Waal asks, when the sound of a drum is heard, is the sound caused by the drum, the drummer, or an interaction between the drum and drummer? Neither the drum nor the drummer in isolation can produce a sound. The sound is a product of an interaction between the nature of the drum and the drummer [25]. This ‘interaction analogy’ can also be applied in a consumer behavior context. That is, consumer behavior is not a product of either one’s nature or environmental factors acting in isolation. But rather, consumer behavior is a product of an interaction between one’s biological nature and one’s environment.

Three lines of research that demonstrate biology–environment interactions and the effect on consumer behavior include the effect of economic cycles on the drive to beautify, the effect of environmental harshness on appetite and food preference, and the effect of early childhood environments on adult consumer behavior.

### 2.4. The Effect of Environmental Conditions on the Drive to Beautify

Economic recessions are associated with high unemployment, lower wages, and resource scarcity [10]. As a result, environmental conditions created by economic recessions can make survival more difficult. Research has shown that recessions are associated with an increase in desire for female personal care and beautifying products [8,11]. Baardwijk and Franses also found that economic recessions are associated with an increase in sales of shorter, more revealing dresses [12].

It is postulated that the increase in desire for personal care and beautifying products during an economic recession is adaptive as the act of beautifying makes one more desirable to potential mates who can help share financial burdens. Acquiring a partner can aid survival through resource sharing and enable the biological drive for reproduction [8]. Beautifying may also increase the probability of acquiring employment in a market where there is a shortage of jobs.

### 2.5. The Effect of Environmental Conditions on Appetite and Food Preference

Perceptions of environmental harshness also affect appetite and the desire for low- and high-energy-dense foods. Specifically, when an environment is perceived to be safe, the desire for low-energy-dense foods such as fruits and vegetables increase, whereas the desire for high-energy-dense foods decrease. In contrast, when the environment is perceived to be harsh, the desire for high-energy-dense foods such as steak and bacon increase, and the desire for low-energy-dense foods decrease [13,14].

Evolutionary psychologists explain that these ‘biology–environment’ interactions can be explained by an evolutionary theory called the Insurance Hypothesis. The Insurance Hypothesis posits that harsh environmental conditions trigger an unconscious drive to overconsume high-energy-dense foods. From an evolutionary perspective, under harsh conditions, the preference for energy-dense food is adaptive because harsh conditions are associated with food scarcity [14]. Thus, the preference for energy-dense food increases the likelihood that one will consume more calories than will be burned in the search for food, thus increasing the probability of survival [15].

The relationship between appetite and environmental harshness is also mediated by the intensity of the stress generated. Acute or high-intensity environmental stressors shut down appetite, whereas mild stressors increase appetite [13,14,16,17]. This same effect has been found with products such as furniture, electronics, and leisure products. That is, economic recessions are associated with higher levels of stress and a decrease in desire for these product categories [8].

### 2.6. The Effect of Early Childhood Environments on Adult Consumer Behavior

In addition to ‘current’ environmental conditions affecting consumer behavior, past environmental conditions experienced during childhood can have a profound effect on adult consumer behavior. Environmental conditions can be viewed as being on a continuum, anchored with *safe and nurturing* on one end, and *harsh and unsupportive* on the opposite end. Unstable childhood environments generate uncertainty, chronic stress, and calibrate the brain to be more impulsive and reactive [26]. These traits become enduring personality dispositions that are carried into adulthood and can affect how one responds to environmental stressors. For example, adults who lived in chronically stressful environments during childhood are more likely to respond to stress with higher impulsivity and become spenders rather than savers. In contrast, adults who were raised in safe and nurturing environments tend to respond to stress with lower impulsivity. In addition, when stressed, those who were raised in a nurturing environment are more likely to become savers, rather than spenders [26].

Swaffield and Gou have also shown that adults who were raised under chronically stressful conditions during childhood are more likely to develop “trait appetite”, which is the desire for food in the absence of an energy deficit. In contrast, adults that were raised in low-stress environments are more likely to develop “state appetite” and desire food only when they experience a physiological energy deficit [27].

## 3. The Current Study

Most research that has examined the effect of biology–environment interactions on consumer behavior used the state of the economy (e.g., recessionary conditions) as a proxy measure for environmental conditions. However, ‘the environment’ is multi-dimensional and interweaves dimensions of social support, perceived physical safety, as well as economic and personal financial conditions [14,27]. Thus, to deepen our understanding of how various environmental stimuli interact with one’s nature, it is necessary to isolate and examine the effect of each independent variable on consumer behavior. Specifically, in this study, we examined the effect of environmental conditions on the desire for products that have symbolic value or are used to conspicuously signal a message to others.

Consumers often purchase these products for their symbolic value and as a means to signal or communicate a message to others [28,29,30,31]. For example, the conspicuous display of products can be used to signal social class and which social groups one wants to be known to be affiliated with [32,33,34,35]. Products can also communicate a desired social identity, such as that one is brave, sexually attractive, or possesses physical or financial power [36]. These messages can play an important role in both intra- and intersexual competition.

Therefore, we conducted an experiment to examine the effects of different types of environmental harshness (i.e., social, economic, and physical safety) on female and male preference for products that signal beauty, wealth, and toughness. More specifically, we make the following two hypotheses:

**H1:** 
*Product desire is affected by a biology–environment interaction, rather than environmental factors acting in isolation.*


**H2:** 
*Different types of environmental harshness (social, financial, and physical safety) have different effects on product desire.*


## 4. Methods

This study includes two separate experiments that examine how different types of environmental harshness affect the desire for signalling products. Both studies use a repeated-measures design. The first study investigated female product preference and the second examined male product preference.

### 4.1. Participants

Participants between the ages of 30 and 50 years old were recruited from across Canada through the Qualtrics online survey service (Table 1). 

This age group was selected as it was felt that the products used in the study experiments would appeal to this market segment. With regard to the sampling method, the sample was not chosen, but rather, the participants were self-selected into participation; specifically, the Qualtrics online crowd sourcing service randomly sent emails to people across Canada that fit the age profile and invited them to participate in the study. Once participants logged into the Qualtrics server, they were assigned to one of the six scenarios in their log-in order.

Each participant was paid a nominal fee (~CAD 1.25) to participate. As alcohol and leather products were options in this experiment, individuals who were opposed to buying these products were excluded from this study. This resulted in an initial sample size of 629 participants (315 females, 314 males).

Lastly, a seven-point Likert scale was used to assess pre-test product desire and post-test change in desire. An inherent limitation of Likert scales is that they are unable to measure decreases in desire when the assigned pre-test score is “1”. Likewise, when a score of “7” (the highest score) is assigned in the pre-test, a post-test increase in desire cannot be measured. Due to this limitation, and to increase the precision of our analysis, participants that had a pre-test score of “1” and a post-test score of “1”, or a pre-test score of “7” and a post-test score of “7” were removed from this analysis. After applying this filter, the final sample size was 344 (197 females, 147 males).

### 4.2. Procedures for Experiment 1: Female Product Desire

Having provided informed consent, participants were provided with a URL to the online experiment. Females in this study were shown 10 product images; 5 images of products that signalled wealth and 5 images of beautifying products (Figure 1). The non-brand-specific images were purchased from online stock photography sites and the images with brand name products were taken from the manufacturer’s catalogue. Original images are available from the authors on request. The images were presented in a different randomized order for each participant. Participants viewed each image and rated it according to the following question which was displayed below the image: “How much would you like to buy this product right now?” Ratings were recorded using a 7-point Likert-type scale (anchored with the descriptors 1 = extremely undesirable and 7 = extremely desirable).

We measured the coefficient of reliability between the variables in the pre- and post-product desire tests using Cronbach’s alpha. For female subjects and *beautifying* products, Cronbach’s alpha for the variables included in pre-test desirability questions (i.e., scale level for each product) was 0.62, while that for variables included in post-test desirability questions was 0.76. For female subjects and *wealth*-signalling products, Cronbach’s alpha for the variables included in pre-test desirability questions (i.e., scale level for each product) was 0.79, while that for variables included in post-test desirability questions was 0.89. Thus, the results on the consistency suggest that the questions used in the study can in general be considered robust.

The participants were then randomly assigned to one of six conditions: safe socially supportive environment (safe social), harsh unsupportive social environmental (harsh social), good economic prospects (safe economic), poor economic prospects (harsh economic), safe physical environment (safe physical safety) or dangerous physical environment (harsh physical safety). Next, the participants were asked to read an accompanying scenario text (see Appendix A). Each scenario text depicted environmental conditions associated with the condition group to which they were assigned. The scenario text concluded with the sentence, “Pause for a moment and think about the story you just read. How would you feel if this really was your situation?” Once the scenario had been read, the participants were asked to rate the same products for a second time. The product images were presented in a new and fully randomized order. Finally, participants completed a background questionnaire to obtain demographic details.

### 4.3. Analysis Procedure for Experiment 1: Female Product Desire

In order to determine whether the products indeed fell into two categories (i.e., beautifying products vs. wealth-signalling products), we conducted an Exploratory Factor Analysis (EFA). The dependent variables were the participants’ ratings for each product. Separate EFAs were conducted for both pre- and post-treatment product ratings. We conducted the EFA using Principal Axis Factoring with direct oblimin as the rotation method, using SPSS 21. The number of factors was determined based on eigenvalues and the scree plots. Products were divided into two categories based on the EFA results.

The teeth whitening product and the inexpensive necklace did not cluster with a single group and therefore were removed from the subsequent analysis. Beautifying products included items that are known to accentuate or draw attention to physical features. For example, red nylons can draw an observer’s eyes towards one’s legs, and high-heeled shoes can make one’s legs look firmer and more attractive [37]. The final items included in the beautifying product set included nail polish, non-brand-name (inexpensive) high-heeled shoes, and red nylons.

The second set of female products that signalled wealth included a Coach^TM^ handbag, leather jacket, Gucci^TM^ shoes, Prada^TM^ sunglasses, and a Pandora^TM^ bracelet. While these expensive products may make a female more appealing, they are less likely to make her ‘sexually’ attractive. Nor are wealth-signalling products likely to be used to accentuate physical features that are desired by men. We can infer that this assumption is correct as the EFA did not include any of the wealth-signalling products in the beautifying product cluster.

### 4.4. Results for Experiment 1: Female Product Desire

We examined whether the environmental condition (safe vs. harsh) and product category (beautifying vs. financial wealth-signalling products) affected participants’ *change* in desire in each scenario (social, financial, physical safety). For each of the three scenarios, we conducted a dummy-variable linear regression analysis, where the dummy variable signalled whether the person was subject to either the safe or harsh environmental conditions.

#### 4.4.1. Female Beauty-Signalling Products

##### The Effect of Social Conditions on Desire for Female Beautifying Products

Neither the perception of socially safe (secure) nor harsh (insecure) conditions moderate desire for female beautifying products. Specifically, for beautifying products and *social* scenarios, the estimation results indicated that for subjects who were given the *safe* (secure) social scenario, their change in desirability was on average −0.088 but not statistically significant (t = −0.746, *p* = 0.458; equivalently, F (1,83) = 0.557, *p* = 0.458), while for those who were given the *harsh* social scenario (insecure conditions), their change in desirability was on average −0.057 and again, not statistically significant (F (1,83) = 0.288; *p* = 0.593).

##### The Effect of Financial Conditions on Desire for Female Beautifying Products

Perceptions of safe (secure) financial conditions did not affect the desire for female beautifying products. In contrast, perceptions of unsafe, harsh financial conditions decreased desire for female beautifying products.

With regard to beautifying products and the *financial* scenarios, the estimation results indicated that for subjects that were given the *safe* (secure) financial scenario, their change in desirability was on average 0.0488 and not statistically significant (t = 0.3, *p* = 0.765; equivalently, F (1,82) = 0.09, *p* = 0.765), while for those that were given the *harsh financial* scenario (insecure financial condition), their change in desirability was on average −1.047 and strongly statistically significant (F (1,82) = 43.537; *p* = 3.812 × 10^−9^).

##### The Effect of Perceived Physical Safety on Desire for Female Beautifying Products

For beautifying products and the *physical safety* scenarios, the estimation results indicated that for subjects who were given the *safe* physical scenario, their change in desirability was on average −0.231 but not statistically significant (t = −1.34, *p* = 0.183; equivalently, F (1,84) = 1.802, *p* = 0.183), while for those who were given the *harsh* (unsafe) physical safety scenario, their change in desirability was more negative on average −0.794 and strongly statistically significant (F (1,84) = 25.727, *p* = 2.316 × 10^−6^). Note, these results are inconsistent with consumer consumption patterns associated with low-intensity environmental stressors, but consistent with consumption patters associated with high-intensity environmental stressors [27] (Figure 2).

#### 4.4.2. Female Wealth-Signalling Products

##### The Effect of Social Conditions on Desire for Female Wealth-Signalling Products

For wealth-signalling products and the *social* scenario, the estimation results indicated that for subjects who were given the *safe* (secure) scenario, their change in desirability was on average −0.342 and marginally significant at the 5.3% level (t = −1.97, *p* = 0.053; equivalently, F (1,66) = 3.8804, *p* = 0.0531), while for those who were given the *harsh* (insecure) scenario, their change in desirability was on average −0.049 but was not statistically significant (F (1,66) = 0.094, *p* = 0.76).

##### The Effect of Financial Conditions on Desire for Female Wealth-Signalling Products

For wealth-signalling products and the *financial* scenario, the estimation results indicated that for subjects who were given the *safe* (secure) scenario, their change in desirability was on average 0.065 and not statistically significant (t = 0.29, *p* = 0.773; equivalently, F (1,77) = 0.084, *p* = 0.773), while for those who were given the *harsh financial* scenario, their change in desirability was on average −1.471 and strongly statistically significant (F (1,63) = 57.511, *p* = 6.419 × 10^−11^).

##### The Effect of Perceived Physical Safety on Desire for Female Wealth-Signalling Products

For wealth-signalling products and the *physical safety* scenario, the estimation results indicated that for subjects who were given the *safe* (secure) scenario, their change in desirability was on average −0.28 and marginally statistically significant at the 7% level (t = −1.88, *p* = 0.063; equivalently, F (1,75) = 3.551, *p* = 0.063), while for those that were given the *harsh* scenario, their change in desirability was on average −1.07 and strongly statistically significant (F (1,75) = 47.987, *p* = 1.271 × 10^−9^).

Note, similar to the results with beautifying products, the decrease in desire for financial wealth-signalling products are inconsistent with consumer consumption patterns associated with low-intensity environmental stressors, but consistent with consumptions patters associated with high-intensity environmental stressors [27] (Figure 3).

### 4.5. Procedure for Experiment 2: Male Product Desire

A second factor analysis was conducted on the products purchased by men. The products shown in Figure 4 were speculated to fit each respective category. However, the biker sunglasses, weights, ring, and Cognac alcohol did not cluster with a single group and therefore were removed from the subsequent analysis. The first distinct cluster included products that are associated with signalling toughness and bravery. This cluster included mountain climbing equipment, an arm tattoo, and a motorcycle. The second product cluster included expensive products that signalled wealth. This cluster included a watch, expensive suit, and a leather jacket (Figure 4).

### 4.6. Analysis for Experiment 2: Male Product Desire

Similar to the female product analysis, we conducted EFAs to determine the number of product categories. We identified two distinct product categories: products that signalled *toughness* and bravery (tattoo, motorcycle, and climbing equipment) and products that signalled *wealth* (classy watch, suit, and brown leather jacket). While we did not test if our study participants perceived these products to be associated with toughness and perceptions of wealth, other studies have. Arm tattoos and motorcycles are more common with the lower socioeconomic classes because they symbolize masculinity and toughness [38,39,40], whereas expensive and classy clothing tends be associated with higher socioeconomic classes [41]. Therefore, we assume that the products are appropriately categorized as ‘toughness-signalling’ and ‘wealth-signalling’ products. Also, it is notable that despite the fact that motorcycles are expensive, the EFA did not cluster the motorcycle with the other expensive products, but rather, it was clustered with the toughness products. Therefore, we assumed that the participants in this study perceived the motorcycle to symbolize toughness rather than wealth. Lastly, participants who had a pre-test score of “1” and a post-test score of “1” or a pre-test score of “7” and a post-test score of “7” were removed from this analysis (same rationale as in the female analysis).

Similar to the female analysis, we measured the coefficient of reliability between the variables in the pre-and post-product desire tests using Cronbach’s alpha and, in general, the results were consistent with those for female subjects. Specifically, for *toughness* products, Cronbach’s alpha for the variables included in pre-test questions was 0.64, while that for variables included in post-test questions was 0.67. Finally, for male subjects and *wealth*-signalling products, Cronbach’s alpha for the variables included in pre-test desirability questions was 0.71, while that for variables included in post-test desirability questions was 0.80. Thus, as before, the results suggest that the questions used in the study can in general be considered robust.

We examined whether the environmental condition (safe vs. harsh) and product category (toughness-signalling vs. financial wealth-signalling products) affected participants’ *change* in desire in each scenario (social, financial, physical safety). For each of the three scenarios, we conducted a dummy-variable linear regression analysis, where the dummy variable signalled whether the person was subject to either the safe or harsh environmental conditions.

#### 4.6.1. Male Toughness-Signalling Products

##### The Effect of Social Conditions on Desire for Male Toughness-Signalling Products

For *toughness* products and the *social* scenario, the estimation results indicated that for subjects who were given the *safe* scenario, their change in desirability was on average 0.099 but not statistically significant (t = 0.638, *p* = 0.53; equivalently, F (1,56) = 0.407, *p* = 0.53), while for those who were given the *harsh* scenario, their change in desirability was on average −0.065 and again, not statistically significant (F (1,56) = 0.199, *p* = 0.66).

##### The Effect of Financial Conditions on Desire for Male Toughness-Signalling Products

With regard to products that are used to signal male toughness and the *financial* scenario, the estimation results indicated that for subjects who were given the *safe* financial scenario, their change in desirability was on average −0.046, but this was not statistically significant (t = −0.195, *p* = 0.85; equivalently, F (1,54) = 0.038, *p* = 0.85), while for those who were given the *harsh* financial scenario, their change in desirability was on average −0.716 and was strongly significant (F (1,54) = 8.616, *p* = 0.005).

##### The Effect of Perceived Physical Safety on Desire for Male Toughness-Signalling Products

For toughness-signalling products and the *physical* safety scenario, the estimation results indicated that for subjects who were given the *safe* physical safety scenario, the change in desirability was not statistically significant (t = −0.734, *p* = 0.47; equivalently average −0.095, F (1,52) = 0.538, *p* = 0.47), while for those who were given the *harsh* physical safety scenario, the change in desirability was on average −0.089 and was also not statistically significant (F (1,52) = 0.443, *p* = 0.51) (Figure 5).

In summary, the only scenario that affected a change in desire for toughness-signalling products was in the harsh financial scenario. One might infer that a decrease in desire makes intuitive sense as harsh financial environmental conditions are often associated with lower income. However, this pattern is more closely associated with adults who were raised in safe (comfortable) socioeconomic environments. Specifically, under harsh financial conditions, those who were raised in lower socioeconomic environments tend to become spenders, whereas consumers raised in upper social class become savers [26].

Also as noted prior, the perceived intensity of the environmental stressor can have a differential effect on product desire. Specifically, in some product categories, low-intensity stressors tend to increase desire, whereas high-intensity stressors tend to decrease desire. In fact, this point is theoretically supported in this study by comparing the effect of financial stressors on females (Experiment 1) relative to males (Experiment 2). Numerous studies have shown that on average, when exposed to the same stressful event, females rate the event as more stressful in comparison to males [42,43].

#### 4.6.2. Male Wealth-Signalling Products

##### The Effect of Social Conditions on Desire for Male Wealth-Signalling Products

For *wealth-signalling* products and the *social* scenario, the estimation results indicated that for subjects who were given the *safe social* scenario, their change in desirability was on average 0.016 but this increase in desirability was not statistically significant (t = 0.18, *p* = 0.86; equivalently, F (1,81) = 0.032, *p* = 0.86). Likewise, those who were exposed to the *harsh social* scenario, their change in desirability was on average −0.139 and again, not statistically significant (F (1,81) = 2.438, *p* = 0.12).

##### The Effect of Financial Conditions on Desire for Male Wealth-Signalling Products

For *wealth-signalling* products and the *financial* scenario, the estimation results indicated that for subjects who were given the *safe financial* scenario, their change in desirability was on average 0.048 but not statistically significant (t = 0.239, *p* = 0.81; equivalently, F (1,85) = 0.057, *p* = 0.81), while for those who were given the *harsh* financial scenario, their change in desirability was on average −1.237 and strongly significant at any conventional level (F (1,85) = 41.27, *p* = 7.3 × 10^−9^).

The Effect of Perceived Physical Safety Conditions on Desire for Male Wealth-Signalling Products: For wealth-signalling products and the physical scenario, the estimation results indicated that for subjects who were given the safe physical safety scenario, their change in desirability was on average −0.109 and not statistically significant (t = −0.832, *p* = 0.41; equivalently, F (1,83) = 0.692, *p* = 0.41), while for those who were given the harsh physical safety scenario, their change in desirability was on average −0.598 and strongly statistically significant (F (1,83) = 17.781, *p* = 6.28 × 10^−5^).

In summary, harsh financial conditions have a similar effect in that it decreases desire for both toughness-signalling products and wealth-signalling products. However, what is different is that harsh physical safety conditions reduce desire for wealth-signalling products, but not toughness-signalling products. This finding reinforces the concept that a decrease in desire for these products may be driven by other factors such as desire to not be noticed, rather than a decrease in income (Figure 6).

## 5. Discussion

The purpose of this study was twofold. First, to examine the relationship between perceptions of environmental harshness (safe/harsh) and product desire. The second purpose was to examine whether product desire was affected by the type of environmental harshness (social, financial, and physical safety). The results empirically support the hypothesis that product desire is affected by a biology–environment interaction, rather than environmental factors acting in isolation. The results also support the hypothesis that different types of environmental harshness have different effects on product desire.

One might ask, how can it be inferred that a biology–environment interaction exists, and the results are not an outcome of environmental stimuli acting in isolation? Two lines of reasoning support this inference.

First, the *harsh* environment scenario stories (the primes) used in the experiments were intended to generate stress and the stress hormone cortisol. In contrast, the *safe* environment scenarios were written in a manner to generate a relaxed, non-stressed feeling. The study results show statistically significant differences in product desire between the safe environment scenarios and the harsh environment scenarios. Therefore, it can be inferred that the harsh environment primes did create a stressed state of mind and elevated cortisol levels. For these reasons, it can be inferred that a biology–environment interaction exists.

A second line of reasoning that supports the existence of a biology–environment interaction is based on prior neurological research that shows that on average, women and men process aversive stimuli differently. Specifically, females respond more strongly to aversive stimuli than males [42,43]. This variation is, in part, a result of biological differences in the amount of stress hormone (cortisol) produced, and differences in how men and women process stressful stimuli. When stressed, men tend to show more activity in their prefrontal cortex than women. In contrast, when stressed, women show more brain activity in their limbic system, which is responsible for processing emotions [44]. As a result, while female and male participants in this study read the same environment scenario stories (the primes), they likely felt different levels of stress, which differentially affected product desire.

As stated earlier in this paper, some consumer behaviors are seen as irrational, especially when they lead to problems such as compulsive buying behavior, overeating, and the accumulation of excessive debt. However, these behaviors may in fact be adaptive and aid in mate attraction, reproduction, and survival. For example, products can be used to signal messages to others. Wealth-signalling products can be used to both attract a mate (a biological need), and to intimidate same sex rivals by making them feel less able to compete. Likewise, in a harsh unsafe environment, products that signal toughness can serve as a form of protection by intimidating potential aggressors [36,45,46]. Lastly, prior research has also shown that harsh environments are often associated with resource scarcity and lower survival rates. In these harsh environments women are also attracted to physically dominant men who have access to scarce resources [39,47,48].

At first glance, it may appear that the results of the two studies included in this paper, contradict prior research that has shown that exposure to harsh environmental conditions increase product desire. That is, the results of this study did not show that perceived environmental harshness increased product desire. But rather, depending on the type of environment scenario, the effect of perceived environmental harshness had either no effect, or a statistically significant negative effect.

A potential explanation for these differences may be due to the intensity of the stress generated by the scenario stories that were used in this study. As previously noted, the intensity of a stressor can mediate and moderate consumer behavior. Low-level chronic stress tends to increase the desire for beautifying products such as miniskirts, lipstick [8,11], and high-energy-dense foods [13]. In contrast, high-intensity or acute stress tends to generate a short-term fight-or-flight survival response that is focused on removing the immediate threat [49]. Thus, it is possible that the level of stress generated from the scenario stories in this study triggered a self-protection motive, which subsequently led to a decrease in product desire. Thus, a less intense stressor may have an opposite effect and increase product desire.

Another factor that may have affected the results is the average age of the participants. The average age of both the female and male participants was 38 years old. As identified earlier, people may desire products because of their signalling value. Wealth-signalling products and beautifying products can make one more attractive to potential mates. However, the biological drive to mate-up and reproduce decreases as one becomes older [50]. Thus, it is plausible that the results may have been different if the participants were in their sexual prime (e.g., mid-twenties). In addition, numerous research studies have shown that under high-stress situations, both female and male sex drives decrease [51,52,53]. Thus, it is possible that the intensity of the harsh environment scenario stories used in this study could lead to a decrease in product desire for both females and males.

A final point that is relevant to both the female and male analyses is that signalling one’s wealth through products in a harsh environment could increase one’s risk of being robbed and physically harmed. Researchers Griskevicius, Ackerman, and Redden state that humans have a self-protection survival drive. If one does not physically survive, they may not have an opportunity to find a mate and reproduce [26]. Thus, it can be assumed that in extremely harsh environments, the self-protection drive will supersede the drive to use products to signal potential mates or potential rivals. In addition, under intensely harsh environment conditions, one is likely to avoid displaying one’s wealth due to the dangers associated with doing so.

## 6. Study Limitations

While this study offers valuable insights into how nature–nurture interactions can influence product desire, there are a number of limitations that may impact the interpretation of the results. Understanding these limitations can also help researchers identify areas for further research.

The first point that should be considered when interpreting the results of this study is that while the need to demonstrate one’s attractiveness, wealth, and toughness may be common across all cultures, the products consumers use to signal these traits may vary. Factors such as age, socioeconomic status, and subculture may also influence the selection of products used to augment attractiveness and to promote a perception of wealth and toughness.

A second consideration is that the products used in this experiment were selected because they are commonly used by consumers to directly or indirectly (e.g., beautifying products) signal a message to others. These products are sometimes referred to as ‘conspicuous consumption’ products. It should be acknowledged that it is unknown whether the findings from this study can be generalized to other products that are consumed or used in a non-conspicuous manner.

A third limitation of this study was the nature of Likert scales. If on a 7-point scale, a participant selects ‘1’ in the pre-test, they cannot rate their desirability lower than a ‘1’ in the post-test; thus, this would not capture a change in desire. Likewise, if a participant rates desirability at a ‘7’ in the pre-test, they cannot rate desirability any higher in the post-test. As a result, when the pre-test score is subtracted from the post-test score, it would appear that there is no change in desirability—which may not be true. A more precise measure of change in desirability could be obtained if researchers were able to first identify a list of products that each participant has a neutral feeling towards (e.g., rated a desirability of 4 on a 7-point Likert scale). By starting from a point of neutrality in the pre-test, researchers would then be able to have a more valid measure of post-test changes in desirability.

Lastly, it is important to remember that this study measured a change in product desire. It is unknown how closely changes in product desire will mirror actual purchase behavior.

## 7. Practical and Theoretical Implications

This research has provided empirical support for the hypothesis that consumer product preferences can be affected by an interaction between biological predispositions and environmental conditions. This finding has implications for both marketing practitioners and academics who conduct theoretical marketing research.

As much is still unknown about the effect of nature–nurture interactions on product preference, marketing and consumer psychology researchers may be wise to acknowledge that these latent interactions represent a potential confounding variable.

A second implication of this research study is that it shows that consumer product preferences may be less stable than is often assumed. This has implications for researchers and marketers who aim to match product preferences with different market segments. Thus, it may be prudent to profile product preferences relative to different environmental conditions.

Additionally, when marketers are assessing market potential and forecasting sales, it would be wise to consider current and future environmental conditions. For example, governments can implement policies that affect how consumers perceive their environment. A relevant example is how governments around the world managed the 2020–2022 COVID-19 epidemic. Government policies prevented social gatherings which created a socially harsh environment and a sense of isolation. They also published regular reports on how many people were dying due to COVID-19. This created a feeling that the environment was physically dangerous. Lastly, many governments implemented policies that prevented businesses from operating. The impact of this policy was that it created financially harsh environments with high unemployment.

The fourth implication of this research is relevant to how marketers evaluate the effectiveness of their marketing and promotional initiatives. Specifically, when assessing the effectiveness of an organization’s marketing efforts, it would be wise to assess if the organization’s level of performance was a product of their initiatives or a nature–nurture interaction that the marketer had no control over. By conducting this analysis, marketers are better able to refine their marketing plans to better meet consumer needs.

## 8. Conclusions

The purpose of this study was to test whether product desire can be influenced by an interaction between one’s biological predispositions or nature, and environmental conditions (nurture). In addition, it was hypothesized that different types of environmental harshness (social, financial, and physical safety environments) have a differential effect on product desire. Through the use of a repeated-measures study design, both hypotheses were empirically supported. Specifically, the data show that with regard to products that are used in a conspicuous consumption manner, as environmental harshness increases, product desire decreases. This is a noteworthy finding because it compliments prior research that shows that the intensity of an environmental stressor can mediate product desire in a nonlinear manner.

This line of research supports the perspective that a fundamental shift is needed in how we think about what drives consumer behavior. The findings from this study have implications for understanding how variations in environmental conditions can impact consumer needs. It also has implications for assessing market potential and forecasting sales. Finally, by analyzing consumer behavior through a biology–environment interaction lens, we are better able to understand the etiology of problematic consumption behaviors which may lead to the discovery of potential remedies.

## Figures and Tables

**Figure 1 behavsci-14-00789-f001:**
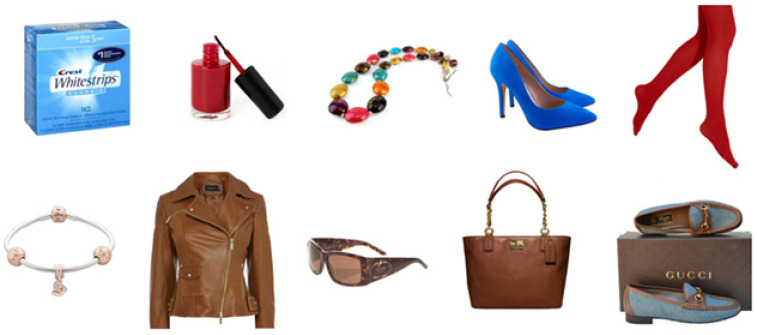
Female products used in experiment. Images were shown individually and in randomized order. Top row (left to right): Beautifying products—* Crest White Strips (teeth whitener), red nail polish, * inexpensive necklace, high-heel shoes, red nylons. Bottom row: wealth signalling products—Pandora^TM^ bracelet, leather jacket, Prada^TM^ sunglasses, Coach^TM^ handbag, Gucci^TM^ shoes. * = products that did not cluster with either beautifying or wealth signalling products and were removed from analysis.

**Figure 2 behavsci-14-00789-f002:**
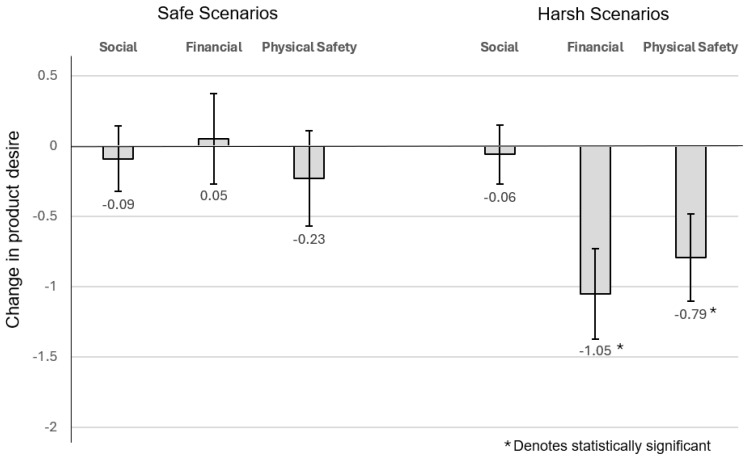
Change in desire for female beautifying products.

**Figure 3 behavsci-14-00789-f003:**
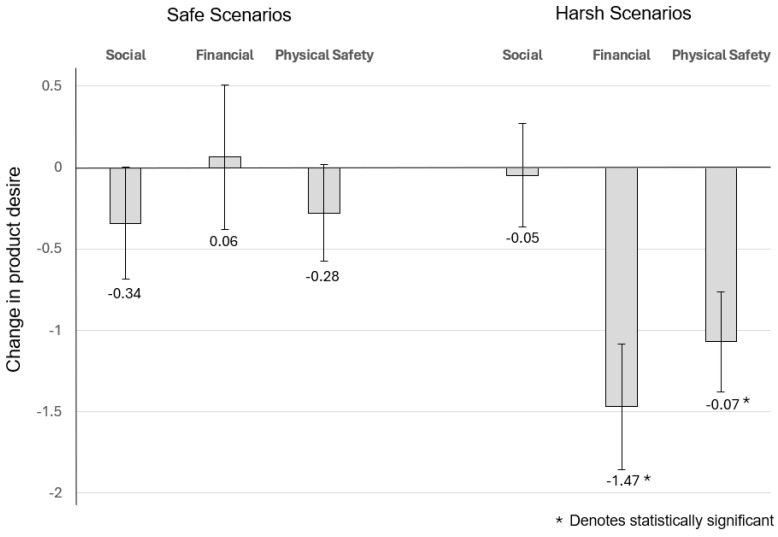
Change in desire for female wealth signalling products.

**Figure 4 behavsci-14-00789-f004:**
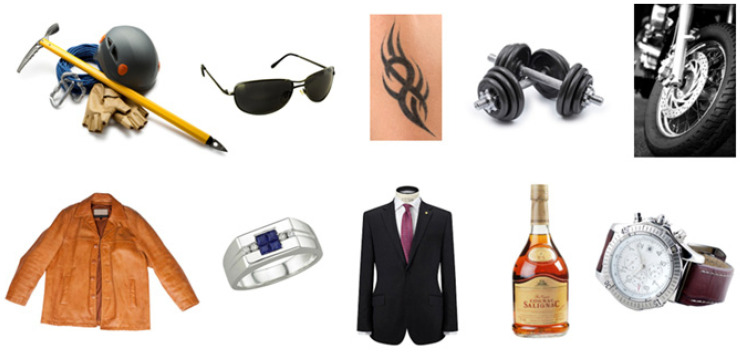
Male products used in experiment. Images were shown individually and in randomized order. Top row (left to right): Toughness signalling products–mountain climbing equipment, * biker sunglasses, arm tattoo, *dumb bells, and motorcycle. Bottom row: Wealth signalling products–leather jacket, men’s ring, suit jacket, *Cognac alcohol, expensive watch. * = products that did not cluster with either beautifying or wealth signalling products and were removed from the final analysis.

**Figure 5 behavsci-14-00789-f005:**
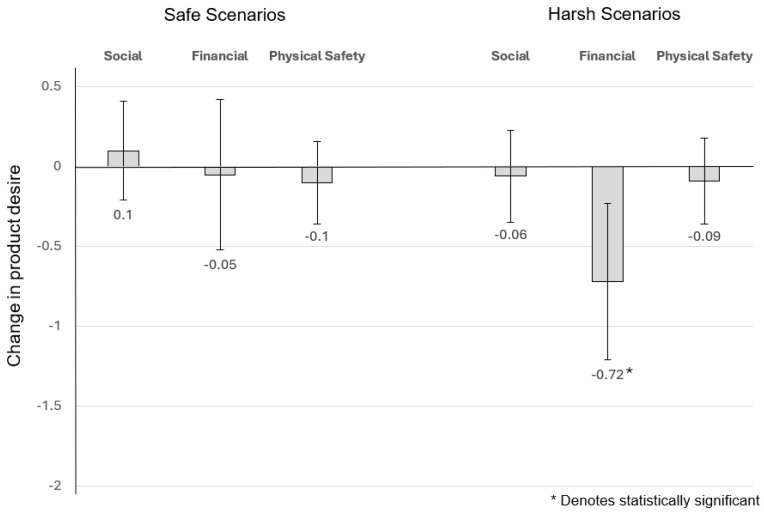
Change in desire for male toughness signalling products.

**Figure 6 behavsci-14-00789-f006:**
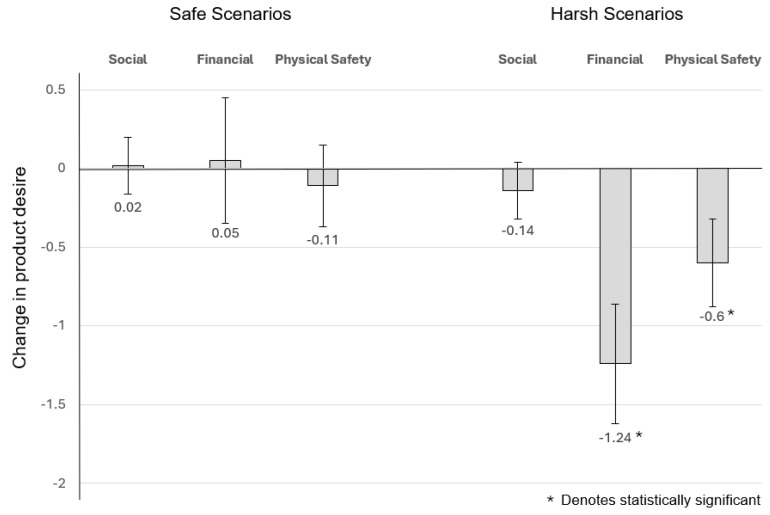
Change in desire for male wealth signalling products.

**Table 1 behavsci-14-00789-t001:** Mean age and sex of participants in each scenario.

		Female		Male	
		Count	Mean Age and s.d.	Count	Mean age and s.d.
Scenario	Safe social	25	39 (5)	23	37 (5)
	Harsh social	35	37 (6)	27	39 (5)
	Safe financial	26	39 (6)	25	38 (7)
	Harsh financial	40	39 (5)	27	39 (6)
	Safe physical	36	39 (5)	27	38 (5)
	Harsh physical	35	38 (5)	18	40 (6)

s.d. refers to standard deviation of the sample.

## Data Availability

The datasets generated during and/or analyzed during the current study are available from the corresponding author on reasonable request.

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
