# Peer review of "Unconscious Drivers of Consumer Behavior: An Examination of the Effect of Nature–Nurture Interactions on Product Desire"

_behavsci, 2024, doi:10.3390/bs14090789_

Round 1

Reviewer 1 Report

Comments and Suggestions for Authors

Thanks for submitting this study; a very interesting title and topic here. Nevertheless, a few matters need to be addressed:

1—While your topic is intriguing, providing a clear justification for your research is essential. Perhaps this could be in the introduction part.  I found it challenging to identify the gap in the literature that your study aims to fill. The gap should be based on previous recent papers in the field; please address these in the introduction and literature review section. 

2- while you've provided a methodological section, it is more like a method you used. You need a proper methodology with clear philosophical underpinnings.

3- Additionally, please clearly state how you ensured the reliability of your findings. e.g. Have you done any pilot?

4- You mention a payment of $1.25 per participant; what measures did you take to make sure that the results were not biased.

3- You stated that you have ethical approval from the Ethics Review Board at the University of Stirling via a letter of approval and by the University of Alberta Ethics Review Board, registration number Pro00043569_REN1. Could you kindly include this approval in your next submission.  

4-Could you kindly include your paper's practical and theoretical implications into separate sections? Preferably state these before and after the conclusion section. 

6-The limitation section could be expanded a little bit more.

Author Response

Reviewer Comment 1: While your topic is intriguing, providing a clear justification for your research is essential. Perhaps this could be in the introduction part.  I found it challenging to identify the gap in the literature that your study aims to fill. The gap should be based on previous recent papers in the field; please address these in the introduction and literature review section. 

Author’s Reply to Comment 1:

We will reply to the above comment regarding “not being able to see a gap in the existing literature,” with reference to Dr. Cheng Lu Wang’s editorial titled, “The misassumptions about contributions,” which was published in the Journal of Interactive Marketing Vol. 16, No. 1.)

In this article Dr. Wang states, “An interesting study is not evaluated by whether it has been studied; it should indicate something important but overlooked (e.g., an important intervening variable, a boundary condition or resolving controversial and contrasting findings), adding new value, new insights and many times counterintuitive and beyond conventional wisdom.”

Dr. Wang continues, “an interesting study would demonstrate what seems to be X, is in reality non-X or what is accepted as X is actually a non-X.”

As noted by consumer psychology y researcher Gad Saad (2014), the traditional way to study consumer behavior is from a Standard Social Sciences Model (SSSM).  A criticism of the SSSM is that this approach tends to examine proximate causes of consumer behavior.

However, our study is breaking new ground by analyzing consumer behavior through an evolutionary psychology lens (nurture-nature interaction perspective) which can lead to an inference about the ultimate causes of behavior.  Simply stated, we are entering unchartered territory where there is a paucity of prior research to build upon. 

We feel that the above paragraph also is relevant to the reviewer’s comment, “I found it challenging to identify the gap in the literature that your study aims to fill. The gap should be based on previous recent papers in the field.” 

In sum, this research study contributes to a paradigm shift in how what we think about what triggers product desire.  Metaphorically, we are not adding to the traditional consumer psychology “wall of  science,” but rather contributing the “evolutionary psychology wall of science.”

Reference:
Saad, G. (2014). The evolutionary bases of consumption. Psychology Press.

Reviewer Comment 2:  While you’ve provided a methodological section, it is more like a method you used. You need a proper methodology with clear philosophical underpinnings.  

Author’s Reply:

We have included both a description of the methodological procedures and the study design (which I think you are looking for).   Specifically, in section 5.2, we included the following paragraph which explains that we have used a “Within-Subjects /Repeated Measures design.”   This is the gold standard in social sciences research. 

Methods section 5.2, it states,

“The participants were then randomly assigned to one of six conditions: safe socially supportive environment (safe social), harsh unsupportive social environmental (harsh social), good economic prospects (safe economic), poor economic prospects (harsh economic), safe physical environment (safe physical safety) or dangerous physical environment (harsh physical safety).  Next, the participants were asked to read an accompanying scenario text (see Supplementary Material).  Each scenario text depicted environmental conditions associated with the condition group to which they were assigned.  The scenario text concluded with the sentence, “Pause for a moment and think about the story you just read. How would you feel if this really was your situation?”  Once the scenario had been read, the participants were asked to rate the same products for a second time.  The product images were presented in a new and fully randomized order.  Finally, participants completed a background questionnaire to obtain demographic details.”

We assume that this will provide the philosophical underpinnings for this methodology.

Reviewer Comment 3 - Additionally, please clearly state how you ensured the reliability of your findings. e.g. Have you done any pilot?

Author’s Reply:

The reliability of the findings was ensured through the following procedures. 

First, the sample size was large.  As noted in ‘Table 1. Mean age and sex of participants in each scenario,’ there was 197 females and 147 males in this study. 

Second, this was an online study that was administered across Canada through the Qualtrics research platform.  Thus, we sampled from a very large geographic region.

Third, as shown by the standard deviation of age (5-6 S.D.’s) in Table 1, we have a wide and representative age range.

Fourth, the male and female participants were randomly assigned to one of the six environmental prime categories.

Fifth, the product items were shown to the participants in a random order in both the pre- and post experimental conditions.

In sum, these five conditions should support the inference that the findings are reliable and generalizable to the larger population both within Canada and internationally.

With regards to running a pilot, we are not sure what type of pilot you are referring to.  The scenario stories (environmental primes) that were used in this study had been used before in a study that was conducted by the principal investigator of this current study.

Reviewer Comment 4 - You mention a payment of $1.25 per participant; what measures did you take to make sure that the results were not biased.

Author’s Reply:
At the time the participants logged into the study, they would not know how much they get paid.  This is because the way that Qualtrics works, is by paying the participants by time (per minute). Thus, the participants could participate in a single study that takes 10 minutes, or in three in a row that amount to 45 minutes.  On average, the participants in this study were paid $1.50 US.

As this is a nominal token amount, we do not believe that the amount paid would bias any results.  Also, the large sample size would also reduce a bias if there was one. 

Reviewer Comment 5 - You stated that you have ethical approval from the Ethics Review Board at the University of Stirling via a letter of approval and by the University of Alberta Ethics Review Board, registration number Pro00043569_REN1. Could you kindly include this approval in your next submission.  

Author’s Reply:

All the requested approval forms have been received by the Behavioral Sciences Assistant Editor Ms. Lana Tucakovic.  She has also confirmed that the ethics approval forms were approved on July 12, 2024.  Ms. Lana Tucakovic can be reached at lana.tucakovic@mdpi.com for confirmation.

Reviewer Comment 6 - Could you kindly include your paper’s practical and theoretical implications into separate sections? Preferably state these before and after the conclusion section. 

Author’s Reply:

This section has been updated. Please see updated transcript.

Reviewer Comment 7 -The limitation section could be expanded a little bit more.

Author’s Reply:

This section has been updated. Please see new transcript.

Reviewer 2 Report

Comments and Suggestions for Authors

The paper investigates the effect of biology-environment interactions on product desire. To accomplish this, the study employs two experiments (i.e., 315 females and 314 men) that used a repeated-measures design. The results showed that under harsh economic conditions, product desire generally decreased.  Based on the paper provided for review, I outline my recommendations.

1-Firstly, it is strongly suggested that the paper provide deeper insights into the literature and carve out what we know, what the theoretical problem in the existing literature is, and how your study can help overcome this problem by offering new and novel generalised insights. Furthermore, the introduction section of the paper does not showcase the research gaps identified from recent literature and the authors should consider propositions on adopting some theoretical foundations for this study.

2-Secondly, a clear articulation of the overview of each section should be provided where a short paragraph before the literature review can improve the overall flow of the paper.

3-Each section of the literature review lacks substantial evidence from recent studies.

4--Within the paper, the author mentions four hypotheses being proposed, but only two hypotheses are present in the subsequent paragraphs. The weakest segment of the paper lies in the methods section, which does not provide much information on how sampling was chosen and what type of sampling method was undertaken in the study. Moreover, no justification is provided for why two specific products (i.e., leather and alcohol) were chosen to examine how different types of environmental harshness affect the desire for signalling products.

5- Further, the study should provide an explanation on whether the sample population is chosen from a specific country, and if not, are the products showcased in the experiment generalized for a wider population? Moreover, it is understood that clothing behaviour and perceptions of wealth-signalling products differ across cultures and various countries. Hence, it would be beneficial to provide a niche context to the study rather than keeping it broad.

6- The discussion sections related to each of the findings lack support from recent studies.

7- The paper does not showcase sections such as practical and academic contributions, which are important parts of a study. Further, the authors should strive to improve the paper's limitations section. Finally, while the paper reads well and has a straightforward structure, the authors' arguments should be supported by recent literature.

Author Response

Reviewer Comment 1- Firstly, it is strongly suggested that the paper provide deeper insights into the literature and carve out what we know, what the theoretical problem in the existing literature is, and how your study can help overcome this problem by offering new and novel generalized insights. Furthermore, the introduction section of the paper does not showcase the research gaps identified from recent literature and the authors should consider propositions on adopting some theoretical foundations for this study.

Author’s Reply:

We will reply to the above comment regarding “not being able to see a gap in the existing literature,” with reference to Dr. Cheng Lu Wang’s editorial titled, “The misassumptions about contributions,” which was published in the Journal of Interactive Marketing Vol. 16, No. 1.)

In this article Dr. Wang states, “An interesting study is not evaluated by whether it has been studied; it should indicate something important but overlooked (e.g., an important intervening variable, a boundary condition or resolving controversial and contrasting findings), adding new value, new insights and many times counterintuitive and beyond conventional wisdom.”

Dr. Wang continues, “an interesting study would demonstrate what seems to be X, is in reality non-X or what is accepted as X is actually a non-X.”

As noted by consumer psychology y researcher Gad Saad (2014), the traditional way to study consumer behavior is from a Standard Social Sciences Model (SSSM).  A criticism of the SSSM is that this approach tends to examine proximate causes of consumer behavior.

However, our study is breaking new ground by analyzing consumer behavior through an evolutionary psychology lens (nurture-nature interaction perspective) which can lead to an inference about the ultimate causes of behavior.  Simply stated, we are entering unchartered territory where there is a paucity of prior research to build upon. 

We feel that the above paragraph also is relevant to the reviewer’s comment, “I found it challenging to identify the gap in the literature that your study aims to fill. The gap should be based on previous recent papers in the field.” 

In sum, this research study contributes to a paradigm shift in how what we think about what triggers product desire.  Metaphorically, we are not adding to the traditional consumer psychology “wall of  science,” but rather contributing the “evolutionary psychology wall of science.”

Reference:
Saad, G. (2014). The evolutionary bases of consumption. Psychology Press.

Reviewer Comment 2  Secondly, a clear articulation of the overview of each section should be provided where a short paragraph before the literature review can improve the overall flow of the paper.

Author’s Reply: We suspect that there may be a disconnect with the focus of this paper.  Specifically, this paper is not really about the desire of certain types of products.  But rather, it is a paper that  is examining whether there is empirical support for the hypothesis that product desire can be mediated and moderated by a nurture-nature interaction.    As noted in comment 1, there is a paucity of research in this area.  We are truly breaking new ground.  Thus, I am not sure how we can edit this section as requested. 

Reviewer Comment 3 -Each section of the literature review lacks substantial evidence from recent studies.

Author’s Reply:

While there are studies that have found interaction effects in their statistical analysis, there are no studies (recent or past) that have looked a “nature-nurture” interactions on product desire.  We are breaking new ground in this area.

Reviewer Comment 4 -Within the paper, the author mentions four hypotheses being proposed, but only two hypotheses are present in the subsequent paragraphs. The weakest segment of the paper lies in the methods section, which does not provide much information on how sampling was chosen and what type of sampling method was undertaken in the study. Moreover, no justification is provided for why two specific products (i.e., leather and alcohol) were chosen to examine how different types of environmental harshness affect the desire for signaling products.

Author’s Reply:

With regards, to four hypothesize being mentioned, thank you for catching this.  We are in discussions with another reviewer and may make a slight modification to the hypothesize.  We will ensure that we are consistent. Thank you.

We have included both a description of the methodological procedures and the study design Specifically, in section 5.2, we included the following paragraph which explains that we have used a “Within-Subjects /Repeated Measures design.”   This is the gold standard in social sciences research. 

Methods section 5.2, it states,

“The participants were then randomly assigned to one of six conditions: safe socially supportive environment (safe social), harsh unsupportive social environmental (harsh social), good economic prospects (safe economic), poor economic prospects (harsh economic), safe physical environment (safe physical safety) or dangerous physical environment (harsh physical safety).  Next, the participants were asked to read an accompanying scenario text (see Supplementary Material).  Each scenario text depicted environmental conditions associated with the condition group to which they were assigned.  The scenario text concluded with the sentence, “Pause for a moment and think about the story you just read. How would you feel if this really was your situation?”  Once the scenario had been read, the participants were asked to rate the same products for a second time.  The product images were presented in a new and fully randomized order.  Finally, participants completed a background questionnaire to obtain demographic details.”

Reviewer Comment 5 - Further, the study should provide an explanation on whether the sample population is chosen from a specific country, and if not, are the products showcased in the experiment generalized for a wider population?

Moreover, it is understood that clothing behaviour and perceptions of wealth-signaling products differ across cultures and various countries. Hence, it would be beneficial to provide a niche context to the study rather than keeping it broad.

Author’s Reply:

With regards to the question is the study chosen from a particular country, Under the title, “5.1 Participants” it states,

“Participants were recruited from across Canada through the Qualtrics online survey service (Table 1).  Each participant was paid a nominal fee (~$1.25) to participate.  As alcohol and leather products were options in this experiment, individuals who were opposed to buying these products were excluded from this study.  This resulted in an initial sample size to 629 participants (315 females, 314 males).” 

With regards to the question would it be beneficial to provide  niche context to the study because their can be cultural variation regarding what types of products consumers value.  The point that you are making is definitely true.  In fact, I will add to your comment and note that consumers will also vary in what products based on age, socioeconomic status and ethnic group within Canada.  They will also vary from one year to the next based on trends.

However, the types of products are not important to this study.  What is important is testing the theory that product desire is affected by a biology-environment interaction, rather than environmental factors acting in isolation (Hypothesis 1), and that different types of environmental harshness have different effects on product desire (Hypothesis 2).  The product categories, of female beatifying products, wealth signaling products and male toughness signaling products simply provide us with some product variation information. 

Also, all cultures have female beatifying products, wealth signaling products and male toughness signaling products.  What is most important is the product category rather than the product item within the category.

Reviewer Comment 6 - The discussion sections related to each of the findings lack support from recent studies.

Author’s Reply

We hope that we are interpreting the phrase, “the findings lack support from recent studies” correctly.  If we are correct in assuming that you are saying that we have not mentioned any similar studies, then yes, you are correct.  However, the reason that there is a lack of support is because there are very few studies that have examine nature-nurture interactions on product desire.  We are breaking new ground here. There are studies that examine product desire from an evolutionary perspective, however, they are simply examining the “ultimate” motivation for wanting a particular type of product.  For example, females desire beautifying products to attract a mate, which enables reproduction.  Other than that, one of the authors of this study, (Jim Swaffield) has three published papers on nature-nurture interactions on food desire and appetite, but that is pretty much all that is available.  We have cited the extant literature.

Reviewer Comment 7 - The paper does not showcase sections such as practical and academic contributions, which are important parts of a study. Further, the authors should strive to improve the paper's limitations section. Finally, while the paper reads well and has a straightforward structure, the authors' arguments should be supported by recent literature.

Author’s Reply:

A section on practical and academic contributions has not been requested by Behavioral Science before.  In fact, I do not believe that it is in their author guidelines.  That being said, we believe that there is material currently within the paper that touches on practical and academic contributions.  I think that perhaps we can move some things around and perhaps add a few more points to build this area up.

Reviewer 3 Report

Comments and Suggestions for Authors

The research is considered to be significant and intriguing from both scientific and practical perspectives. 

However, there are a few inquiries for the authors:

1) how was the sample's representativeness verified?

2) is utilizing a Likert scale alone adequate for research purposes?

3) can result validity be maintained without the incorporation of neurotechnology? The Limitations section should be elaborated on in relations to these issues.

It is advised recommended to present the results in the Conclusions section that align with the of the study's hypotheses. 

Author Response

Reviewer Comment 1 how was the sample's representativeness verified?

Author’s Reply:

The samples representativeness was verified as follows:

Under the title, “5.1 Participants “ it states, “Participants were recruited from across Canada through the Qualtrics online survey service (Table 1).  Each participant was paid a nominal fee (~$1.25) to participate.  As alcohol and leather products were options in this experiment, individuals who were opposed to buying these products were excluded from this study.  This resulted in an initial sample size to 629 participants (315 females, 314 males).”

The reliability of the findings was ensured through the following procedures. 

First, the sample size was large.  As noted in ‘Table 1. Mean age and sex of participants in each scenario,’ there was 197 females and 147 males in this study. 

Second, this was an online study that was administered across Canada through the Qualtrics research platform.  Thus, we sampled from a very large geographic region.

Third, as shown by the standard deviation of age (5-6 S.D.’s) in Table 1, we have a wide and representative age range.

Fourth, the male and female participants were randomly assigned to one of the six environmental prime categories.

Fifth, the product items were shown to the participants in a random order in both the pre- and post experimental conditions.

In sum, these five conditions should support the inference that the findings are reliable and generalizable to the larger population both within Canada and internationally.

Reviewer Comment 2 - is utilizing a Likert scale alone adequate for research purposes?

Author’s Reply:  You raise a valid point.  There are limitations with using a Likert scale. 

First, as noted in the study, “Under 5.1 Participants “ it states, “Lastly, a seven-point Likert scale was used to assess pre-test product desire and post-test change in desire.  An inherent limitation of Likert scales is that they are unable to measure decreases in desire when the assigned pre-test score is a “1.” Likewise, when a score of “7” (the highest score) is assigned in the pre-test, a post-test increase in desire cannot be measured.  Due to this limitation, and to increase the precision of our analysis, participants that had a pre-test score of “1” and a post-test score of”1,” or a pre-test score of “7” and a post-test score of “7” were removed from this analysis.   After applying this filter, the final sample size was 344 (197 females, 147 males).”

A second limitation of using Likert scales is that they may be insufficient to measure slight changes in desire.  Alternatively, one could use biometric measures such as galvanic skin response, changes in heart rate and pupil dilation, or neurological activity in certain parts of the brain.  However, biomarkers also come with their own problems.  For example, if heart rate increases, we don’t know for certain if the participant likes or dislikes what they see.  The same problem occurs with measuring neurological changes.  Unfortunately, all methods have limitations. 

Reviewer Comment 3 - can result validity be maintained without the incorporation of neurotechnology? The Limitations section should be elaborated on in relations to these issues.

Author’s Reply:

We believe that you are referring to the use of neuromarketing techniques that measure changes in neurological activity within the brain.  Neuromarketing suffers from numerous problems.  For example, neuromarketing cannot measure intent, nor desire. As noted in the previous comment, we can see brain activity (e.g., the limbic system).  However, the limbic system is responsible for fight, flight, freeze, feed and fornicate drives (the 5F’s).  Neuromarketing cannot tell us what response is being measured. It can only tell us that there is neurological activity happening in the brain.

As the field of neuromarketing is still in it’s infancy and is not a limitation, we do not feel that it is necessary to bring it into the conversation as a limitation.

Reviewer Comment 4 - It is advised recommended to present the results in the Conclusions section that align with the of the study's hypotheses.

Author’s Reply:

The results relative to the stated hypotheses have been discussed in section “6. Discussion” of the paper.  As there is a fair amount of detail that is discussed, it may make more sense to keep it in the discussion section.  However, that being said, we are currently addressing  another reviewer’s comment about how the current hypothesize are written.  Once that issue has been resolved, we will see if we can add more to the conclusion section.

Reviewer 4 Report

Comments and Suggestions for Authors

The article is written on an urgent topic and is interesting for a reader. It analyses consumers’ behavior in different conditions of the environment using well-structured experiments. The authors raise the problem of forming conscious consumer choices of goods on the market and try to discover fundamental factors influencing their behavior. This article appeals to me because of its primary research results, the additional material with excellent presentation of the scenarios of stories used in experiments, its hypothesis testing, and the creative approach to choosing objects for analysis in the experiment.

However, the manuscript has some drawbacks:

1)    I recommend adding more keywords in the Abstract and avoiding those already presented in the title (consumer behavior, product desire).

2)    Check the distances between sentences. I found that often, they are too long.

3)    Line 190. The authors mention four hypotheses, but I only see two in the text.

4)    Line 202. I advise enriching this subchapter with some additional material with explanations about the choice of the instrument for gathering the answers of respondents, the method of research, and the demographic characteristics of respondents (such as geographical and educational level; they could be essential while transcribing their answers about their good preferences).

5)    Add an explanation of s.d. in Table 1.

6) The authors mention healthy consumer behavior several times, and even in the Conclusion, they write about it. What do the authors mean by this concept? What is “healthy consumer behavior” in their line of research? I understand that the mainstream of their study is to find ways to increase healthy consumption and reduce unhealthy behavior. Then it is pretty interesting to hear the explanation of why they decided to analyze consumer behavior on these types of goods (like clothing, beautifying goods, and accessories) and not on the example of food, particularly healthy food. The results of this research could not be noticeable enough to draw further conclusions about people's healthy behavior. Otherwise, do not mention healthy behavior in the text of this article, and explore this concept in future papers.

Author Response

Reviewer Comment 1 - I recommend adding more keywords in the Abstract and avoiding those already presented in the title (consumer behavior, product desire).

Author’s Reply: We have edited the keywords. The new key words include “product signaling, environmental harshness, nature-nurture interaction, adaptive behavior.”

Reviewer Comment 2 - Check the distances between sentences. I found that often, they are too long.

Author’s Reply: We are not sure if you are referring to the article being double spaced or the distance between the end of one sentence and the beginning of the next.  Based on my previous publication with the journal Behavioral Sciences, my experience is that these edits are made by the journal once the article is prepared for publication.  We are comfortable with whatever formatting rules Behavioral Sciences uses. 

Reviewer Comment 3 - Line 190. The authors mention four hypotheses, but I only see two in the text.

Author’s Reply: Thank you for catching this.  We are in discussions with another reviewer and may make a slight modification to the hypothesize.  We will ensure that we are consistent. Thank you.

Reviewer Comment 4 - Line 202. I advise enriching this subchapter with some additional material with explanations about the choice of the instrument for gathering the answers of respondents, the method of research, and the demographic characteristics of respondents (such as geographical and educational level; they could be essential while transcribing their answers about their good preferences).

Author’s Reply: with regards to the choice of instrument for gathering respondent answers.  We are not sure if you are referring the use of the Qualtrics research platform or the use of the Likert scale for rating product preference.  Therefore, we will provide an answer for both.

With regards to the use of the Qualtrics research platform, this is an excellent platform as it enables the collection of data across a very large geographic region (e.g., all of North America).  That being said, for this study, we collected data across all of Canada. 

With regards to using a Likert scale for measuring changes in product preference, while Likert scales do have drawbacks (as mentioned in the paper 5.1 Participants – just above Table 1) , other methods such as the use of biomarkers (heart rate, galvanic skin responses, cortisol tests and pupil dilation test) also have their limitations as they cannot tell us if the change in the biomarker represents a change in desire.  

Also, as this was a study that was conducted online, we were stuck with using a Likert scale to collect changes in product desire.

With regards to adding demographic information and geographic location, we agree that the more information that we can collect, the better.  This study was administered across Canada which gives us a great cross-section of consumers from different socio-economic classes, education levels, and so on.   

However, to conduct a meaningful statistical analysis we need at least 30 participants in each group to make any type of statistical inference.  As a starting point, we have six different scenarios, then if we were to subdivide each scenario based on education level, marital status, socio-economic status and so on, our sample size would have to be in thousands. 

Unfortunately, we were not able to have such a large sample size. 

Reviewer Comment 5 - Add an explanation of s.d. in Table 1.

Author’s Reply:  We infer that you are asking that we make a note of what s.d. stands for.  Is this correct?  If yes, we have added the following note below the table: “s.d. refers to standard deviation of the sample.”

Reviewer Comment 6 -  The authors mention healthy consumer behavior several times, and even in the Conclusion, they write about it. What do the authors mean by this concept? What is “healthy consumer behavior” in their line of research? I understand that the mainstream of their study is to find ways to increase healthy consumption and reduce unhealthy behavior. Then it is pretty interesting to hear the explanation of why they decided to analyze consumer behavior on these types of goods (like clothing, beautifying goods, and accessories) and not on the example of food, particularly healthy food. The results of this research could not be noticeable enough to draw further conclusions about people's healthy behavior. Otherwise, do not mention healthy behavior in the text of this article, and explore this concept in future papers.

Author’s Reply:

In the clinical psychology field, the current philosophical approach is not to define unhealthy behavior as an absolute state, but rather from a relative perspective.  For example, if a person is driven to buy or over consume despite their wishes to not do so, then this is a problem for the individual. Thus, an unhealthy behavior is any behavior that creates problems for an individual.   For example, if a person is spending $2000 a month on coffee (which they cannot afford), then this is a problem/unhealthy behavior for them.  However, if a very rich person such as Elon Musk is spending $2000 a month on coffee, and this does not create hardship for the individual, then it is not considered unhealthy behavior.   

In addition, an evolutionary psychology perspective postulates that in some environments the behaviors that we deem to be unhealthy today, are in fact, healthy in a different environment.

see: https://papers.ssrn.com/sol3/papers.cfm?abstract_id=4389902).

Now that being said, the reference to unhealthy behavior in the current paper under review is simply an example of the implications of this line of research and the research findings. The main focus of this study is to test the hypothesis that product desire can be an outcome of a nature-nurture interaction, and to test the hypothesis that different types of environmental harshness can have a different effect. We could have run the study with products that lead to healthy consumption behaviors – which the principle investigator of this paper has already done in the food realm.  In 2015, the principle investigator of this paper ran a study that showed safe (non-stressful) environmental conditions increase appetite for low energy dense foods (fruits and vegetables) and decrease appetite for high energy dense foods (bacon and steak)
see:  https://www.jimswaffield.com/wpcontent/uploads/2017/11/2014_Exposure_to_Cues_of_Harsh_or_Safe_Environmental_Conditions_Alters_Food_Pref.pdf)

In 2020, a second study was run that showed high-intensity environmental stressors decreased appetite for both low and high energy dense foods

see:
https://www.jimswaffield.com/wp-content/uploads/2020/09/Swaffield-and-Guo-Environmental-stress-effects-on-appetite.pdf

Round 2

Reviewer 1 Report

Comments and Suggestions for Authors

Dear Authors, 

Thanks for addressing the comments. 

Author Response

Thank you for your review.  

Reviewer 2 Report

Comments and Suggestions for Authors

The paper investigates the effect of biology-environment interactions on product desire. To accomplish this, the study employs two experiments (i.e., 315 females and 314 men) that used a repeated-measures design. The results showed that under harsh economic conditions, product desire generally decreased.  Based on the paper provided for review, I outline my recommendations and a brief on my comments which would allow the paper to be improved in the upcoming versions.   1- My earlier comments were to provide insights into the literature and carve out what we know, what the theoretical problem in the existing literature is, and how your study can help overcome this problem by offering new and novel generalised insights. Furthermore, the introduction section of the paper does not showcase the research gaps identified from recent literature and the authors should consider propositions on adopting some theoretical foundations for this study.   To address this comment, the authors have indicated that the study represents a paradigm shift in our understanding of what triggers product desire. Metaphorically speaking, we are not just adding to the traditional "wall of science" in consumer psychology, but rather contributing to the "wall of science" in evolutionary psychology.   However, studies focusing on the interplay between nature and nurture, as well as the inclination towards signaling products, should be included to enhance the introductory sections.

  • Pinker, Steven. 2004. Why nature & nurture won't go away. Daedalus 133(4): 5-17
  • Tabery, J. (2023). Beyond versus: The struggle to understand the interaction of nature and nurture. MIT Press.
  • Fuentes, H., Vera‐Martinez, J. and Kolbe, D., 2023. The role of intangible attributes of luxury brands for signalling status: A systematic literature review. International Journal of Consumer Studies47(6), pp.2747-2766.
  • Wan, A., Mundel, J. and Yang, J., 2024. Impulsive and Compulsive Buying and Consumer Well-Being. In Fostering Consumer Well-Being: Theory, Evidence, and Policy (pp. 315-331). Cham: Springer Nature Switzerland.
  • Heilman, R.M., Kusev, P., Miclea, M., Teal, J., Martin, R., Passanisi, A. and Pace, U., 2021. Are impulsive decisions always irrational? An experimental investigation of impulsive decisions in the domains of gains and losses. International Journal of Environmental Research and Public Health18(16), p.8518.

2- Secondly, the authors should consider providing a concise overview of each section, particularly a brief paragraph preceding the literature review (that is, the final paragraph of the introduction), can enhance the overall flow of the paper.    For example- The research paper is made up of XXX sections. The introduction informs the reader of the focus and aim of the effect of biology-environment interactions on product desire. Secondly, the literature review section offers substantive ideas on ..............................., followed by the objectives of the research. Section three describes the methodology of the research and the data-gathering approach. Next, the findings.....................   3-Each section of the literature review lacks substantial evidence from recent studies.    The author claims that the study is a breaking new ground in this area. However, the section "The Effect of Nature-Nurture Interactions on Consumer Behavior" needs refinement with references to arguments mentioned in the sub sections.   4-The weakest segment of the paper lies in the methods section, which does not provide much information on how sampling was chosen and what type of sampling method was undertaken in the study. Moreover, no justification is provided for why two specific products (i.e., leather and alcohol) were chosen to examine how different types of environmental harshness affect the desire for signaling products.   The authors failed to adequately address the comments. The author should include a statement on which sampling method (i.e., Simple random sampling/Stratified sampling/Purposive sampling/ non-probability sampling) was chosen and why it fits the current study/experimentation.   5- Please also try to seek ideas from studies that would provide justification on why two specific products (i.e., leather and alcohol) were chosen to examine how different types of environmental harshness affect the desire for signaling products.   Examples of Studies -

  • Han, Y.J., Nunes, J.C. and Drèze, X., 2010. Signaling status with luxury goods: The role of brand prominence. Journal of marketing74(4), pp.15-30.
  • Schroeder, J.E. and Borgerson, J.L., 2014. Dark desires: Fetishism, ontology, and representation in contemporary advertising. In Sex in advertising (pp. 65-87). Routledge.
  • Pozharliev, R., Verbeke, W., De Angelis, M., Van Den Bos, R. and Peverini, P., 2021. Consumer self-reported and testosterone responses to advertising of luxury goods in social context. Italian Journal of Marketing2021, pp.103-127.

5- Lastly, the authors have mentioned that selecting a 7-Point Likert Scale is a limitation of the study. However, it would be beneficial to include the Cronbach's Alpha Scores for each scale in the findings section to demonstrate the robustness of the scales.

Author Response

Please see the attached PDF.  Thank you for reviewing and considering our manuscript.  Best regards, JS & JS

Reviewer 4 Report

Comments and Suggestions for Authors

The article could be accepted in its present form.

Author Response

Thank you for your review.  Sincerely, JS & JS

Round 3

Reviewer 2 Report

Comments and Suggestions for Authors

The paper investigates the effect of biology-environment interactions on product desire. Additionally, the results showcased that under harsh economic conditions, product desire generally decreased. First and foremost, I would like to thank the authors for considering my suggestions. However, I am still concerned with several aspects and will provide more in-depth insights into how these suggestions could be incorporated to strengthen the paper for publication in Behavioural Sciences. Moreover, please try to incorporate these changes in the paper itself or provide justification in the relevant attachments.

1- Firstly, the authors have stated in their response that "Specifically, we submitted our manuscript for consideration to this special edition precisely because of its relatively greater emphasis on the “managerial implications and practical applications of psychological principles to business practices,” and because its “focus is on real-world decision making and practical implications, and less so on traditional academic frameworks".

Response-

1-My earlier comments were to improve the introduction section and showcase what has been done in this field and later on argue that some of the aspects of consumer behaviour are understudied and needs attention/ missing. Ex-"Although, studies have viewed “nature-nurture interaction from various perspectives but it appears the aspects of XXX and YYY and ZZZ remain understudied. Hence, based on the existing gap within the recent literature, the study investigates on AAA, BBB and its influence/relationship on CCC" .

 Regardless of whether a paper appeals to practitioners or academics, the authors should clarify the context of the study and indicate whether it addresses a country-specific viewpoint or a globally relevant issue. Additionally, this should be supported by key statistics and information explaining the necessity of such a study in a specific country (i.e., Canada).

Rather than just replying to comments, please seek ideas on how to structure an introduction section, as well as from relevant articles that have been published within this special issue.

Example- 

  • Chen, Y. and Zhuang, J., 2024. Trend conformity behavior of luxury fashion products for Chinese consumers in the social media age: Drivers and underlying mechanisms. Behavioral Sciences14(7), p.521.
  • Song, S., Tian, M., Fan, Q. and Zhang, Y., 2024. Temporal Landmarks and Nostalgic Consumption: The Role of the Need to Belong. Behavioral Sciences14(2), p.123.

2-My previous comments included a list of articles that could support the introduction and its justifications. Despite this, after multiple cues, the authors have indicated that these may not be appropriate. If that's the case, the authors should make sure the claims in the introduction are supported by citations from recent research.

 Introduction

"These behaviors are often seen as irrational, especially when they lead to problems such as compulsive buying behavior, overeating, and the accumulation of excessive debt. Unfortunately, many interventions aimed at reducing problematic consumer behaviors have poor long-term effectiveness. For example, training and counselling services that promote weight loss and healthy eating behavior seldom have a positive long-term effect [2]. Debt counselling services and interventions aimed at abating compulsive buying behavior also have a poor record of long-term effectiveness [3, 4]".

3- The weakest segment of the paper lies in the methods section, which does not provide much information on how sampling was chosen and what type of sampling method was undertaken in the study.”

The authors have still not addressed the comments properly and have not provided in the paper a justification on why a certain methodology was utilised in the study. Further, the methodology also does not highlight if a pilot study was conducted or not. Moreover, there is no information on descriptive statistics that was related to the study (Ex- What was income level/marital status/educational status/).

4-Moreover, no justification is provided for why two specific products (i.e., leather and alcohol) were chosen to examine how different types of environmental harshness affect the desire for signaling products. The authors failed to adequately address the comments. 

The authors failed to incorporate this comment in the manuscript. Furthermore, Sections 4, 5.1, 5.2, 5.3, and 5.4 make few references to studies that explain how a particular product can be referred to as beautifying products versus wealth signalling products.

Please also try to seek ideas from studies that would provide justification on why two specific products (i.e., leather and alcohol) were chosen to examine how different types of environmental harshness affect the desire for signaling products.

 Examples of Studies -

  • Han, Y.J., Nunes, J.C. and Drèze, X., 2010. Signaling status with luxury goods: The role of brand prominence. Journal of marketing74(4), pp.15-30.
  • Schroeder, J.E. and Borgerson, J.L., 2014. Dark desires: Fetishism, ontology, and representation in contemporary advertising. In Sex in advertising (pp. 65-87). Routledge.
  • Pozharliev, R., Verbeke, W., De Angelis, M., Van Den Bos, R. and Peverini, P., 2021. Consumer self-reported and testosterone responses to advertising of luxury goods in social context. Italian Journal of Marketing2021, pp.103-127.

5- Lastly, the authors have not addressed my earlier comments. 

"The authors have mentioned that selecting a 7-Point Likert Scale is a limitation of the study. However, it would be beneficial to include the Cronbach's Alpha Scores for each scale in the findings section to demonstrate the robustness of the scales".

Author Response

Dear Reviewer, thank you for your efforts in making our paper better.  We have made some substantial edits to the paper including providing a Cronbach's alpha for both the female and male product studies.  We have also provided more information on the sampling methodology. There are numerous other edits that have been outlined in the attached PDF.

Thank you again,  sincerely, Jim S.
